# Synergy between RecBCD subunits is essential for efficient DNA unwinding

Rani Zananiri[1], Omri Malik[1,2], Sergei Rudnizky[1], Vera Gaydar[1],
Roman Kreiserman[1,3], Arnon Henn[1], Ariel Kaplan[1]*

[1]Faculty of Biology, Technion – Israel Institute of Technology, Haifa, Israel; [2]Russell Berrie Nanotechnology Institute, Technion – Israel Institute of Technology, Haifa, Israel; [3]Faculty of Physics, Technion – Israel Institute of Technology, Haifa, Israel

**Abstract** The subunits of the bacterial RecBCD act in coordination, rapidly and processively unwinding DNA at the site of a double strand break. RecBCD is able to displace DNA-binding proteins, suggesting that it generates high forces, but the specific role of each subunit in the force generation is unclear. Here, we present a novel optical tweezers assay that allows monitoring the activity of RecBCD's individual subunits, when they are part of an intact full complex. We show that RecBCD and its subunits are able to generate forces up to 25–40 pN without a significant effect on their velocity. Moreover, the isolated RecD translocates fast but is a weak helicase with limited processivity. Experiments at a broad range of [ATP] and forces suggest that RecD unwinds DNA as a Brownian ratchet, rectified by ATP binding, and that the presence of the other subunits shifts the ratchet equilibrium towards the post-translocation state.
DOI: https://doi.org/10.7554/eLife.40836.001

*For correspondence:
akaplanz@technion.ac.il

**Competing interests:** The authors declare that no competing interests exist.

## Introduction

The homologous recombination (HR) pathway, responsible for the repair of double-strand breaks (DSBs), is essential to preserve the integrity of the genome and exists in all domains of life (*Sung and Klein, 2006*). In *Escherichia coli*, the HR process is initiated by RecBCD, which binds to the damage site and unwinds the DNA, in preparation for strand invasion (*Dillingham and Kowalczykowski, 2008*). RecBCD possesses three subunits, two of which are super-family one helicases: RecB, a 134 kDa, 3'–5' helicase and RecD, a 67 kDa helicase with 5'–3' polarity. The RecC subunit is a 128 kDa, catalytically dead helicase-nuclease sharing a similar structure to RecB, which acts as a scaffold protein stapling RecB and RecD (*Dillingham and Kowalczykowski, 2008*), and is responsible for recognition of the regulatory Chi sequence (*Handa et al., 2012*; *Singleton et al., 2004*). Previous studies (*Churchill et al., 1999*; *Dillingham et al., 2003*; *Farah and Smith, 1997*; *Kowalczykowski et al., 1994*; *Smith, 2012*; *Spies et al., 2003*; *Wiktor et al., 2018*) have resulted in a working model for the initiation process of HR. First, RecBCD binds the blunt end DNA resulting from the DSB and unwinds the DNA, primarily nicking the 3' tail. The opposing translocation polarities of RecB and RecD allow them to simultaneously translocate on opposite strands of the DNA (*Dillingham et al., 2003*; *Taylor and Smith, 2003*) but, since RecD is faster than RecB under physiological conditions (*Dillingham and Kowalczykowski, 2008*; *Spies et al., 2007*; *Taylor and Smith, 2003*), a single-stranded loop accumulates on the 3'-ended strand (*Taylor and Smith, 2003*; *Wong et al., 2006*). A major conformational change in RecBCD takes place upon recognition of the Chi sequence by the RecC subunit (*Spies et al., 2007*; *Taylor et al., 2014*), which switches RecB's nuclease activity (*Wang et al., 2000*) from nicking primarily the 3'-ended strand to nicking primarily the 5'-ended strand (*Anderson and Kowalczykowski, 1997*; *Dixon and Kowalczykowski, 1993*). Finally, RecA is recruited to the 3'-end of the ssDNA. The RecA nucleofilament is then used to invade the acceptor DNA.

Bulk and single molecule biochemical and biophysical studies of RecBCD revealed that it possesses a remarkable unwinding rate approaching ~ 1600 bp s$^{-1}$ (*Handa et al., 2005*; *Liu et al., 2013*; *Lucius et al., 2002*; *Roman and Kowalczykowski, 1989a*; *Xie et al., 2013*), and real-time imaging of live *E. coli* cells has recently confirmed this result (*Wiktor et al., 2018*). In a recent work, we showed that this fast rate is supported by additional, weak ATP-binding sites, beyond the catalytic ones in RecB and RecD, that serve as auxiliary sites from which ATP is transferred to the catalytic sites. This mechanism was shown to result in a significant increase in RecBCD's unwinding velocity at intermediate ATP concentrations (*Zananiri et al., 2017*). Other works, using ATPase deficient mutants in both subunits, showed that fully active helicase subunits are also required to achieve the fast rates of the complex (*Xie et al., 2013*), suggesting an important role for interactions between the subunits. This is further strengthened by the overall structural organization of RecBCD. The crystal structure of RecBCD's initiation complex shows that 4–6 bps of DNA are unwound at the 'pin' domain of RecC after which the split strands are directed towards the helicase subunits, RecB and RecD (*Singleton et al., 2004*; *Wilkinson et al., 2016*). As a result, one model of RecBCD's translocation postulates that the helicase subunits pull the nascent DNA strands against the pin to catalyze unwinding. Moreover, it was proposed that there is a separation between helicase translocation and unwinding, whereby RecBCD unwinds 4–6 bp of DNA using the pin in a distinct step, and only then the helicase subunits pull the unwound DNA (*Lohman and Fazio, 2018*). In addition, an 'arm' domain of RecB interacts with the DNA 12 bps ahead (*Saikrishnan et al., 2008*; *Singleton et al., 2004*). It was suggested to direct the DNA ends toward the helicase, acting as a guide for the duplex DNA during translocation, to mediate the large translocation step across ssDNA gaps, or to play a direct role in destabilizing the duplex ahead of the translocating enzyme (*Bianco and Kowalczykowski, 2000*; *Dillingham and Kowalczykowski, 2008*; *Simon et al., 2016*). Nonetheless, since the DNA encounters these two structural domains before reaching the helicase domains in RecB and RecD, it was proposed that they play roles in destabilizing the fork (*Lohman and Fazio, 2018*; *Simon et al., 2016*; *Wu et al., 2012*; *Xie et al., 2013*), but their role in the unwinding reaction is still unclear.

Motivated by the fact that long stretches of naked DNA are rare in vivo, the ability of RecBCD to overcome DNA binding proteins 'roadblocks' was studied in vitro (*Finkelstein et al., 2010*). In these studies, it was shown that RecBCD is able to evict or displace bound proteins or complexes, such as RNA polymerase, EcoRI, the lac repressor and even nucleosomes. RecBCD did not pause during these collisions and often pushed proteins thousands of base-pairs before evicting them from DNA. A recent study addressing the physiologically relevant case of multiple roadblocks, showed that RecBCD is able to push the proximal one into the neighboring one, which results in its eviction from the DNA (*Terakawa et al., 2017*). Interestingly, it is likely that the ability to displace these 'roadblocks' is a reflection of the high forces generated by RecBCD. For example, single-molecule studies using DNA unzipping to probe DNA-bound proteins have shown that ~ 20–30 pN of force are required to evict RNA polymerase (*Jin et al., 2010*), nucleosomes (*Hall et al., 2009*; *Rudnizky et al., 2016*; ), and the Egr-1 transcription factor (*Rudnizky et al., 2018*). However, a previous single-molecule study reported that RecBCD is unable to translocate against forces higher than ~8 pN (*Perkins et al., 2004*), although these measurements were performed at very low ATP concentrations (<15 μM). Hence, addressing the mechanism of force generation by RecBCD, and the specific roles played by its subunits, at physiological ATP concentrations, is of great interest.

Here, we present a novel optical tweezers assay that allows us to monitor the activity of RecBCD's individual subunits in the context of a native, full complex. Using this assay, we study the activity of RecD, and compare it with the activity of an isolated RecD subunit, probed using a classical single-molecule helicase assay. Our results show that RecD necessitates structural features present in the other subunits to achieve its full processivity and velocity. Next, by repeating these experiments at a broad range of [ATP] and forces, and comparing the results with the prediction of different kinetic models, we determined that RecD likely functions as a Brownian ratchet, where [ATP] binding traps forward translocation fluctuations to produce net directional movement. Remarkably, our data reveals that structural elements in RecC or RecB shift the ratchet equilibrium towards the post-translocation state, thus stimulating unwinding.

## Results

### Optical tweezers can monitor the force response of individual subunits in a native, full RecBCD complex

Single-molecule experiments using optical or magnetic tweezers have proven to be very useful for studying the mechanism of unwinding by helicases (*Dumont et al., 2006*; *Johnson et al., 2007*; *Lionnet et al., 2007*; *Manosas et al., 2010*; *Manosas et al., 2013*; *Qi et al., 2013*). In these experiments, a DNA or RNA hairpin is held under tension by attaching it to molecular 'handles' that are bound to two trapped microscopic beads (or a single trapped bead and a fixed attachment point). The construct is then exposed to the helicase under study, which can bind the 'fork' of the DNA hairpin and unwind the DNA, resulting in a measurable extension change. Unfortunately, this assay is not appropriate for the study of RecBCD since, in line with its biological function that requires binding to DSB sites, RecBCD's affinity towards DNA lacking a double stranded end is $10^6$-fold weaker than that for blunt ends or short overhangs (*Bianco et al., 2001*; *Roman and Kowalczykowski, 1989a*; *Taylor and Smith, 1985*). Hence, to monitor unwinding and translocation by RecBCD we developed a novel single molecule assay based on a dual trap optical tweezers setup. A DNA construct consisting of a stem with a blunt end mimicking a dsDNA break at one end, is attached to two dsDNA 'tracks', each containing a specific tag at its 5' end, that enable binding to two specifically modified microscopic beads (*Figure 1a*). Each of the beads is trapped in a separate optical trap, thus allowing to apply tension on the construct and monitor its extension. The tether is then moved into a different channel of the laminar flow cell that contains ATP (2 mM, unless specified otherwise) and RecBCD (25 nM). The enzyme binds the blunt end of the DNA and translocates on the stem, without affecting the extension of the tether, until it reaches the fork. Then, due to the opposite polarities of RecB and RecD, each of the subunits translocates on an opposite track, as evidenced by a decrease in the tether's extension (*Figure 1b*) and an increase in the tension (*Figure 1c*). As the force increases beyond 42–50 pN, RecBCD dissociates from the track, resulting in breaking of the tether (*Figure 1b–c*). Control experiments where the stem's end was blocked by ligating a short loop or in the absence of ATP showed no translocation activity. Remarkably, this setup enables us also to study the activity of the individual subunits of RecBCD: If one track is made very short (35 nt; *Figure 1d*), the subunit translocating on it reaches the bead in a very short time, and only the activity of the second subunit, translocating on a long track (4200 bp), affects the tether's extension. Hence, by asymmetrically manipulating the tracks' length, we can directly measure the activity of the individual subunits in the WT RecBCD complex without the need for mutations. Moreover, the applied force directly opposes translocation by each of the subunits as it unwinds its track (*Figure 1—figure supplement 1*), allowing us to study the subunits' mechanical response. *Figure 1e* shows representative traces probing the activities of both subunits (left), RecD (middle), and RecB (right) at four ATP concentrations.

Interestingly, a previous work showed that when the unwinding and translocation of a surface-attached RecBCD was probed against an applied force, direction reversals were observed (*Perkins et al., 2004*). We did not observe these events in our traces for RecBCD or its subunits. Moreover, it was reported that RecBCD is unable to function against forces larger than ~6 pN, in stark contrast to the high forces measured in our experiments (*Perkins et al., 2004*). Since the authors of the previous work ruled out the potential effects of the protein biotinylation or surface interactions, it is possible that these effects are evident only for longer runs of activity and at the low [ATP] concentrations used in their work in order to achieve high resolution.

Finally, since the traces in our experiments are terminated by the force-induced dissociation of the complex from DNA, the observed lengths of the unwinding traces (490 ± 13 bp, 116 ± 16 bp and 380 ± 43 bp, mean ± s.e.m, for RecBCD, RecD and RecB, respectively) do not represent the processivity of the enzyme or its subunits.

### Synergy between its subunits supports unwinding by RecBCD

We use in our experiments a passive mode of operation, where the position of the optical traps is constant and, as the distance decreases between the beads, the force against which RecBCD or a subunit has to translocate increases (*Figure 1b–c*). Hence, we probe the velocity against a range of forces in each individual experimental trace (Materials and methods). We first measured force-

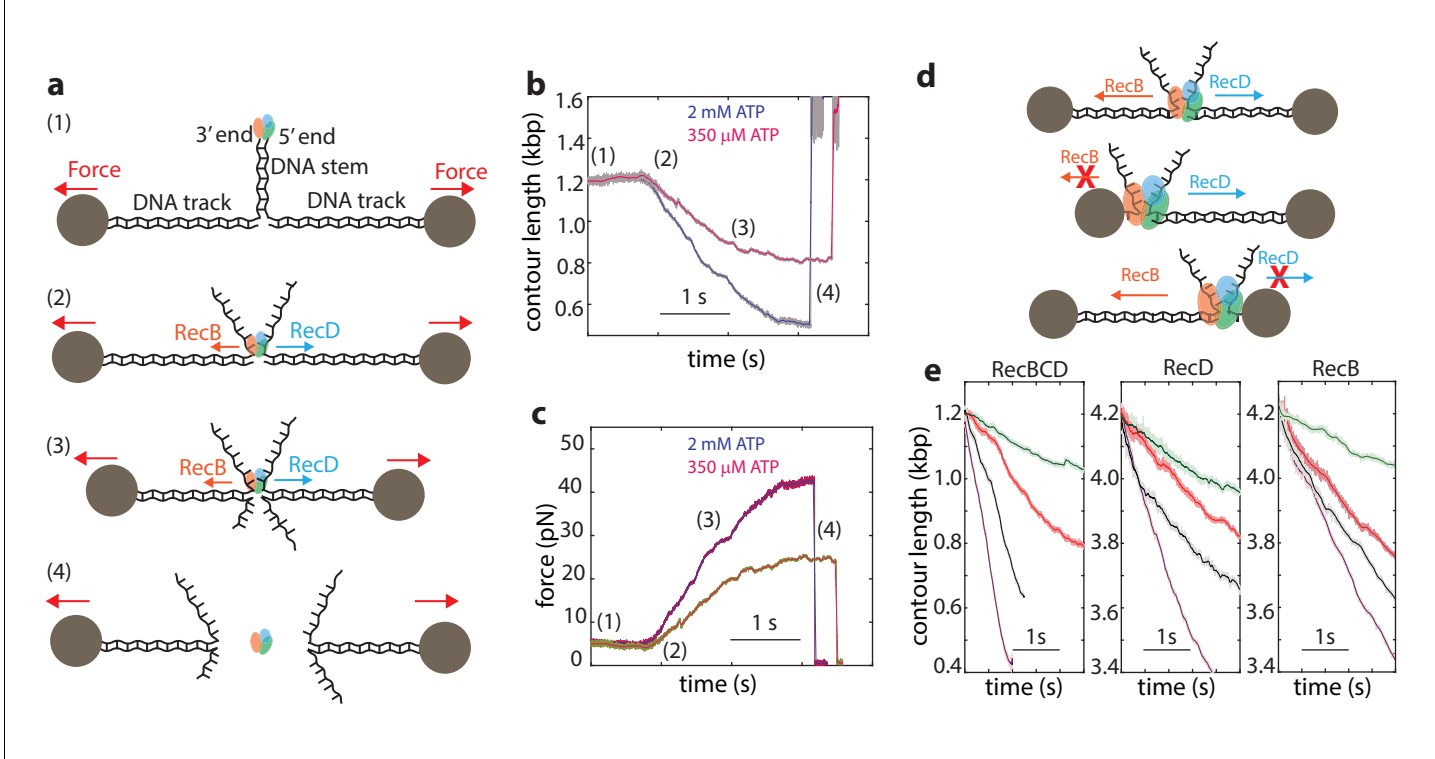

**Figure 1.** Monitoring RecBCD and its individual subunits 'in complex'. (a) Schematic representation of the experimental optical tweezers setup. From top to bottom: RecBCD binds to, and translocates on a DNA stem connected to optically trapped beads through DNA 'tracks'. Upon reaching the fork, the helicase subunits translocate in different directions due to their opposing polarities, shortening the tether length and increasing the tension on it. The force increases up to a point where RecBCD dissociates from the construct. (b) Two representative contour-time traces, for [ATP] = 2 mM (blue) and [ATP] = 350 uM (purple). Raw data is shown in grey, filtered data ($f_c$ = 250 Hz) is shown in blue and purple. (c) The corresponding force-time trace. Raw data is shown in grey; filtered data is shown in blue. (d) Symmetric molecular construct (top, 600 bp and 600 bp) to probe RecBCD and asymmetric constructs (middle, 35 nt and 4200 bp, and bottom, 4200 bp and 35 nt) to probe RecB and RecD, respectively. (e) Representative traces probing both translocases (left), RecD (middle) and RecD (right), at different ATP concentrations (purple, 2 mM; gray, 1 mM; red, 100 µM; green, 20 µM). Raw data is shown in light colors, filtered data ($f_c$ = 250 Hz) is shown in dark colors.

DOI: https://doi.org/10.7554/eLife.40836.002

The following figure supplements are available for figure 1:

**Figure supplement 1.** Schematic description of the force effect.

DOI: https://doi.org/10.7554/eLife.40836.003

**Figure supplement 2.** Linearity of the optical traps at large forces.

DOI: https://doi.org/10.7554/eLife.40836.004

velocity curves under saturating ATP concentrations, for both RecBCD and its subunits. As expected for our experimental geometry, in which the helicase translocates against an opposing force, the velocities of RecBCD and its individual subunits decrease due to the external force (*Figure 2*). However, the drop in the velocity takes place only above a relatively large range of forces at which the opposing force has no effect. This indicates that, in this range, the velocity is not limited by the force-sensitive translocation step, but rather by a force-independent chemical step in RecBCD's mechano-chemical reaction cycle. At higher forces, translocation becomes rate-limiting and results in slowing down RecBCD and its subunits. The force at which the velocity drops to half its maximal value is an indication of the maximal force the enzyme can generate. Hence, *Figure 2a* indicates that RecBCD is able to generate a high force (43 ± 2 pN; Fitted force for half-maximal velocity ± standard error). Considering the typical forces required to displace DNA-bound proteins, as measured by unzipping the DNA (*Jiang et al., 2005*; *Koch and Wang, 2003*; *Koch et al., 2002*; *Meng et al., 2017*; *Rudnizky et al., 2018*) or by using a second DNA molecule as a probe (*Noom et al., 2007*), our measurements provide a mechanical basis for the ability of RecBCD to overcome protein 'roadblocks' (*Finkelstein et al., 2010*; *Gorman and Greene, 2008*;

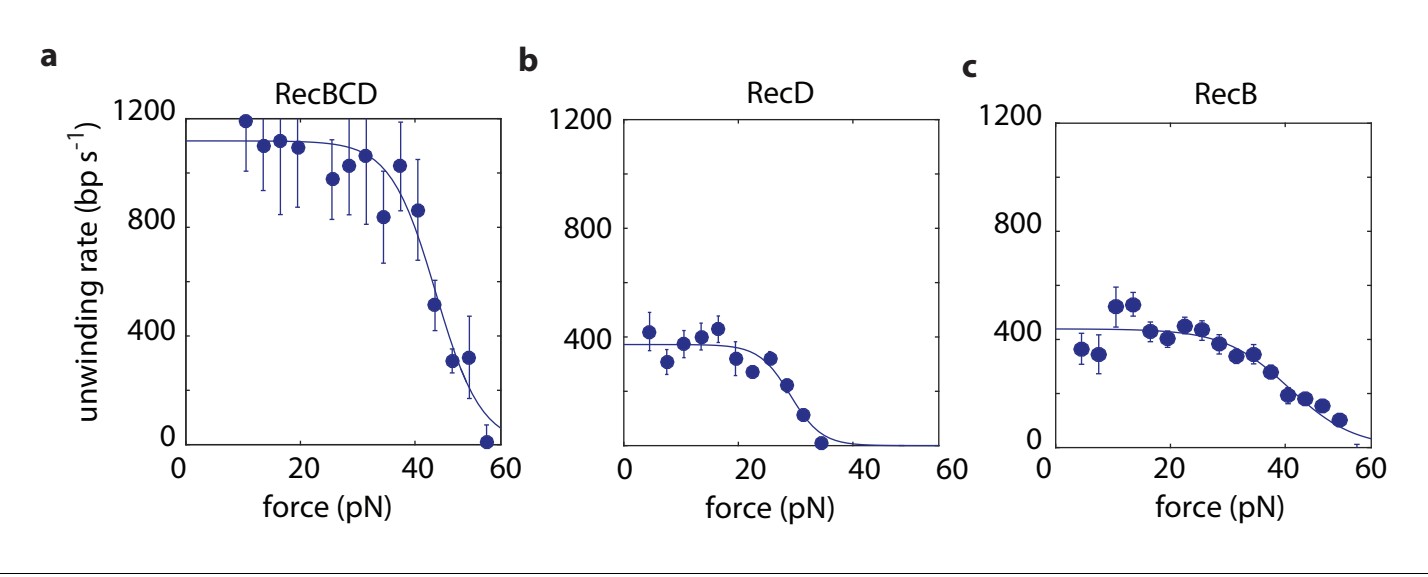

**Figure 2.** Force-velocity curves for RecBCD and its subunits in complex at 2 mM ATP. Force velocity curves at [ATP] = 2 mM for (a) RecBCD, (b) RecD and (c) RecB. Data shown as mean ± s.e.m.; the number of experiments is listed in *Supplementary file 4*. Lines through the data are best fits to *Equation 1*.

DOI: https://doi.org/10.7554/eLife.40836.005

The following figure supplement is available for figure 2:

**Figure supplement 1.** The secondary translocase of RecB does not affect the force-velocity measurements of RecD.

DOI: https://doi.org/10.7554/eLife.40836.006

*Terakawa et al., 2017*). Interestingly, although RecB is able to generate similar forces (40 ± 3 pN; *Figure 2c*), the maximal forces generated by RecD are ~ 30% smaller (29 ± 3 pN; *Figure 2b*), suggesting that this subunit, by itself, is unable to generate the maximal forces observed for RecBCD.

Recent studies have shown that RecB possesses a 'secondary' translocase activity (*Wu et al., 2010*; *Wu et al., 2012*; *Xie et al., 2013*), which translocates on DNA in the 5'→3' direction, that is with an opposite polarity with respect to its 'primary' translocase. Hence, the activity attributed above to RecD may, in principle, be the result of both RecD and RecB's secondary translocase. To test whether the secondary translocase contributes to the activity observed in our measurements, we expressed and purified the mutant RecB$^{K29Q}$CD, which is deficient in RecB's ATPase activity and was shown to be inactive for both translocases (primary and secondary) of RecB (*Xie et al., 2013*). If the secondary translocase affects our measurements, we would expect a different behavior between RecBCD and RecB$^{K29Q}$CD on the asymmetric construct designed to probe RecD. However, *Figure 2—figure supplement 1a* shows that the measured force-velocity curves are indistinguishable. In addition, we measured the activity of the RecBC complex, lacking the RecD subunit. Although we were able to follow its activity in our assay (*Figure 2—figure supplement 1b*), our measurements show that it can sustain maximal forces of only ~9 pN's, suggesting that the secondary translocase is not able to maintain its grip of the DNA, and therefore not active, at higher forces. Taken together, this indicates that our measurements are indeed able to probe separately the activities of RecB and RecD as they are part of a full complex.

At low forces, experiments with the asymmetric constructs reveal that both subunits unwind DNA at ~400 bp/s. Experiments probing the whole complex, where each subunit unwinds a different track, show an unwinding rate of ~1100 bp/s, significantly higher than the sum of the rates measured for the individual subunits. This is in line with previous experiments showing that, while each of the subunits (RecB, RecD) is capable of independently catalyzing unwinding, neither of the subunits alone can reach the unwinding rates of RecBCD (*Dillingham et al., 2005*; *Pavankumar et al., 2010*; *Taylor and Smith, 2003*; *Xie et al., 2013*). The 2–3 fold higher velocity previously measured for RecBCD relative to its subunits indicates some degree of synergy between them. Our results, which

show a 1.4-fold faster activity when both subunits are present but unwinding different substrates, suggest that at least part of this effect results from direct allosteric interactions between the subunits.

Our data shows that DNA is efficiently unwound by the activity of the RecD subunit when it is part of the full RecBCD complex. Therefore, to clarify whether the rest of the complex supports the activity of RecD, it is essential to examine whether RecD as an isolated subunit can also support unwinding. Hence, we expressed and purified the RecD protein (Materials and methods) to compare its force-velocity behavior with that of RecD in complex. Notably, since the experimental configuration used so far takes advantage of the two opposite translocations of the subunits, each one taking place on a different strand of the DNA, it cannot be used to test an isolated subunit. Thus, we used a different experimental geometry (*Figure 3a*), as was previously used for other helicases and polymerases (see for example Refs. (*Dumont et al., 2006*; *Johnson et al., 2007*; *Lionnet et al., 2007*; *Malik et al., 2017b*; *Malik et al., 2017a*; *Manosas et al., 2010*; *Manosas et al., 2012*; *Manosas et al., 2013*; *Morin et al., 2012*; *Qi et al., 2013*)). Here, a DNA hairpin is held under tension between two handles attached to beads trapped in optical tweezers. Upon introduction, the helicase unwinds the hairpin, increasing the extension of the tether and reducing the tension on the hairpin.

The contour-time data for RecD showed multiple activity peaks per trace (*Figure 3—figure supplement 1*). Each of these peaks, when zoomed in, showed a gradual increase in the extension of the tether, followed by a sudden drop (*Figure 3b* and *Figure 3—figure supplement 1*). We interpret this as multiple unwinding events that are terminated by dissociation of the enzyme. RecD showed poor processivity (~ 50 bps), in accordance with previous measurements (*Dillingham and Kowalczykowski, 2008*). When compared with the reported ~ 30,000 bp processivity of

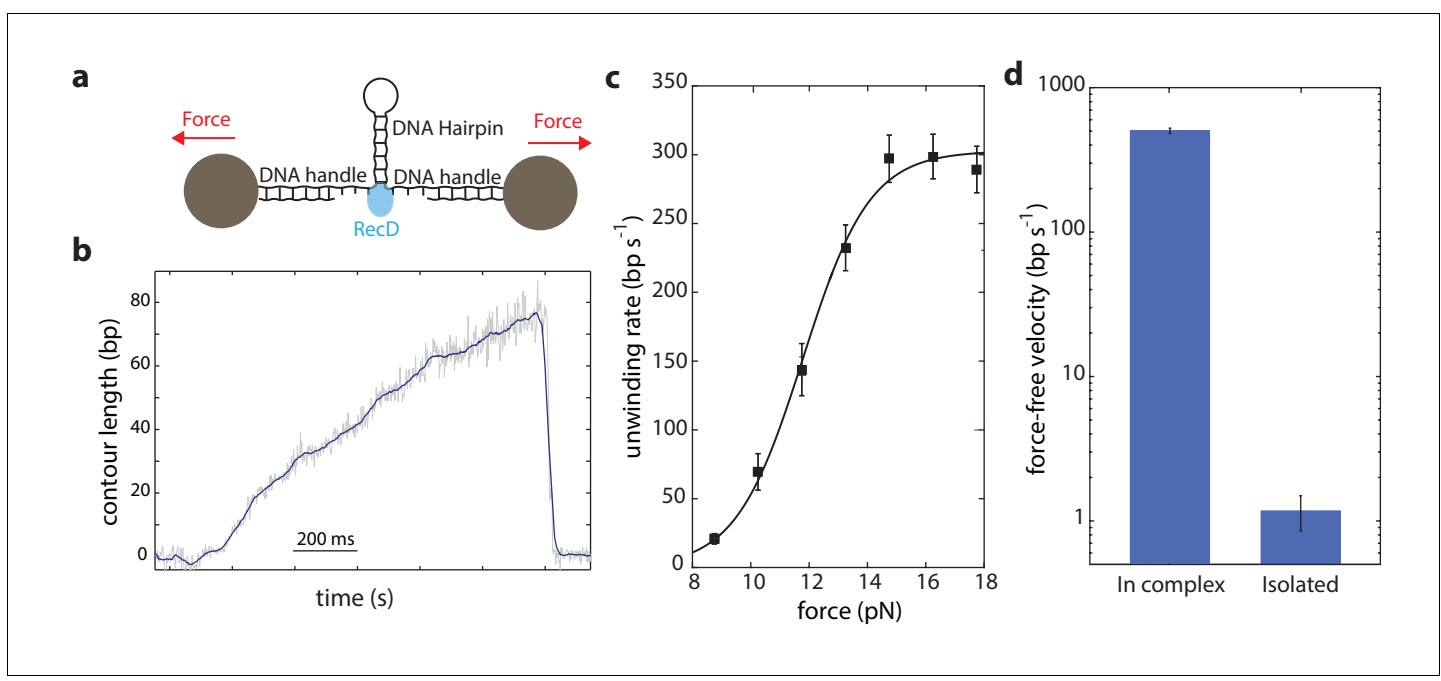

**Figure 3.** Unwinding of a hairpin under tension by the isolated RecD. (a) Schematic representation of RecD unwinding in optical tweezers. (b) Representative unwinding trace. Raw data is shown in grey, filtered data ($f_c$ = 250 Hz) is shown in blue. [ATP] = 2 mM. (c) Velocity-force curve for the isolated RecD at 2 mM ATP follows a sigmoidal-like behavior saturating at high forces. Data shown as mean ± s.e.m., number of traces in *Supplementary file 4*. The line through the data is a best fit to Equation S29, *Supplementary file 3*. (d) Force-free velocities for RecD, obtained from (c) and *Figure 2b* by extrapolating to F = 0, show ~ 2–3 orders of magnitude difference between the enzyme in-complex and the isolated one (Extrapolated result ± s.e.).

DOI: https://doi.org/10.7554/eLife.40836.007

The following figure supplement is available for figure 3:

**Figure supplement 1.** Multiple unwinding events by RecD occur on the same DNA molecular construct.
DOI: https://doi.org/10.7554/eLife.40836.008

RecBCD (*Dillingham and Kowalczykowski, 2008*), this indicates that the presence of RecC and/or RecB, perhaps acting as a 'toehold' as previously proposed (*Carter et al., 2016*), is required to fully achieve RecBCD's processivity.

As opposed to the previous setup (*Figure 1a*), here the force acts as an aiding factor for the helicase and thus, as the helicase unwinds and the force drops, the velocity decreases (*Figure 3a*). The force-velocity curve shows a logistic shape with a plateau at high forces (*Figure 3c*) indicating that when the fork is significantly destabilized, RecD's activity is mainly limited by a chemical step in its mechano-chemical cycle. However, at low forces, RecD's velocity dropped significantly indicating that RecD is severely limited in its ability to disrupt the dsDNA fork, that is that the isolated RecD is a weak helicase. Remarkably, although the two experimental configurations used (*Figure 1a* and *Figure 3a*) are different, their extrapolation to zero force reflects the same physical situation (i. e. unwinding of an unperturbed dsDNA), and therefore are expected to exhibit the same unwinding rates. However, comparing RecD's activity in both configurations, extrapolated to zero force, reveals a significant difference (*Figure 3d*): while RecD in complex can unwind and translocate at very fast rates (~430 bp/s), the isolated subunit is practically unable to unwind the fork. Thus, we conclude that the presence of the RecC and/or RecB subunits is necessary for the RecD subunit to efficiently and processively unwind DNA.

## RecBC stimulates RecD unwinding, by shifting the translocation equilibrium

Mechanistically, the facilitating action of RecB/RecC on RecD may be achieved by different means, such as allosterically increasing the rate of ATP binding, hydrolysis or product release. Therefore, to further clarify the mechanism by which the other subunits in the complex support unwinding by RecD, we analyzed the mechano-chemical cycle of RecD, and the expected effect of an external force in the two sets of experiments presented above, characterizing the unwinding of RecD in complex and the isolated RecD. The ATPase cycle of the helicase can be simplified to the following minimal three step reaction, described by ATP binding (with forward and reverse rate rates $k_{+b}$ and $k_{-b}$, respectively), irreversible hydrolysis ($k_h$) and product release ($k_r$) steps:

$$R + T \underset{k_{-b}}{\overset{k_{+b}}{\rightleftharpoons}} R \cdot T \overset{k_h}{\rightarrow} R \cdot D \cdot Pi \overset{k_r}{\rightarrow} R + D + Pi \, ,$$

where $R$ is the helicase-DNA complex, $T$ is ATP, $D$ is ADP and $Pi$ is inorganic phosphate. In addition, for a processive helicase performing multiple enzymatic cycles without dissociation, the complete kinetic cycle must include also a translocation step, at which the helicase moves forward by a single step ($R_n \rightarrow R_{n+1}$, where $n$ denotes the number of steps carried out by the helicase on the DNA substrate). To characterize how this translocation step is incorporated into the enzyme's ATPase cycle, two questions need to be addressed (*Malik et al., 2017a*): First, it is necessary to determine the *location* of the translocation step within the chemical steps comprising the ATPase cycle. For instance, the translocation step for the XPD helicase takes place after ATP binding (*Qi et al., 2013*). In addition, the *mechanism* by which translocation takes place should be elucidated, between two alternative ideal mechanisms: a 'power stroke' (PS) mechanism postulates that the energy resulting from a chemical step is directly harnessed to power translocation, making the chemical reaction and the translocation tightly coupled. (Note, that the power stroke *mechanism* should not be confused with the large conformational change that powers the movement, also denoted 'power stroke' in the molecular motors community). For a PS mechanism, the translocation is integrated with the ATPase cycle by considering a specific chemical step as involving also the translocation. In contrast, a 'Brownian ratchet' (BR) mechanism postulates that the enzyme thermally and rapidly fluctuates between the pre- and post-translocation state, and that these fluctuations are converted into directional movement by a chemical step that 'traps' the post-translocation configuration. Hence, a BR mechanism is integrated by including an additional step in the cycle, before the rectifying chemical step. Taken together, this means we can postulate six models for the Mechano-chemical cycle, as described in the *Supplementary file 1*. All these models predict an [ATP]-dependent unwinding rate ($v$) that follows Michaelis-Menten kinetics, $v = v_{max}([T]/(K_M + [T]))$, but have different predictions for the steady state kinetic parameters $v_{max}$ and $K_M$ as a function of the applied force.

Notably, each of our experimental geometries (*Figure 1a* and *Figure 3a*), will be affected differently by the force. In the assay of *Figure 1a*, when RecD is active as part of the whole complex, the

external force directly opposes translocation. Assuming that the translocation step is a simple, single barrier crossing step, the forward and backwards kinetic rates are modulated by the factors $\exp(-Fx^{\ddagger}/k_bT)$ and $\exp(F(\delta - x^{\ddagger})/k_bT)$, respectively, where $x^{\ddagger}$ is the distance to the forward translocation transition state and $\delta$ is the step size. Alternatively, in the configuration of *Figure 3a*, used to probe RecD in isolation, there is no external force on the helicase. However, the force on the tether affects the helicase velocity by modulating the stability of the unwinding fork: since translocation depends on the existence of an open fork, and DNA breathing fluctuations are very fast (*Betterton and Jülicher, 2005*), the open fork can be considered as a 'substrate' of the translocation reaction. Therefore, a step that involves forward translocation will be modulated by the factor $P_{open}$, equal to the probability of finding an open fork that is at least the size of the helicase step (*Malik et al., 2017a*). $P_{open}$ is a function of the inherent stability of the DNA fork, a potential destabilization energy provided by the helicase, and the destabilization by the applied force. . Finally, we used the force-dependence of the translocation step, in each of the experimental geometries, to derive equations for the force dependence of the Michaelis-Menten parameters, for each of the models in *Supplementary file 1*, and for each of the two experimental geometries (*Supplementary file 2* and *3*). With these equations in hand, and given a set of model-dependent parameters, it is possible to calculate the velocity predicted by each one of the models, for any force and [ATP]. Of note, the existence of auxiliary ATP-binding sites as we recently showed (*Zananiri et al., 2017*) does not affect the interpretation of the force dependence of the Michaelis-Menten parameters, as it can be accommodated by considering *effective* ATP binding and unbinding rates.

To elucidate which of the models best describes RecD's mechanism of unwinding, we globally fitted the expressions derived above to the experimental data, taken at a broad range of forces and [ATP]. First, we performed a global fitting of each one of the expressions in *Supplementary file 2* and *3* to both experimental datasets simultaneously, that is fitting the same model, with the same microscopic rates ($k_{+b}$, $k_{-b}$, $k_{+h}$ and, for BR models, $k_{tr}$ and $k_{tr}$). However, this procedure failed to recapitulate the force dependence of the Michaelis-Menten curves in both experimental setups. This is not surprising, since we showed in the previous section that RecD relies on a contribution of RecC and/or RecB for disrupting the fork. We assume that the presence of the other subunits is less likely to alter the reaction pathway by which RecD functions, but rather affect (e.g. allosterically) the microscopic *rates* of the biochemical transitions. Therefore, we fitted each one of the models to each one of the two datasets separately, that is using the same model to fit both datasets but allowing for different microscopic rates for each case. This strategy was successful in recapitulating the data, revealing that the model that best describes RecD, in complex and isolated, is a BR where ATP-binding traps the system in the post-translocation state (*Figure 4*, *Figure 4-figure supplements 1* and *2*). Interestingly, a previous single-molecule study reported on extensive conformational dynamics of the RecBCD-DNA complex, arising from back-and-forth motion of the enzyme relative dsDNA (*Carter et al., 2016*). These dynamics were observed in the absence of ATP and depend on the GC content of the DNA, but are suppressed by a non-hydrolyzable ATP analog. Given that RecD is the leading helicase before the Chi sequence (*Liu et al., 2013*; *Spies et al., 2007*) and in view of our results, it is possible that the previously observed dynamics represent the fluctuations of RecD between its pre- and post-translocation states.

Remarkably, comparing the rate constants resulting from the global fitting, for RecD in complex and when isolated, revealed that most microscopic rates are not greatly affected, and that the only profound effect of being in complex is reflected in shifting the equilibrium of the ratchet towards the post translocation state, by approximately 200-fold (*Figure 5*). Interestingly, two structural features in the RecBCD complex, which have been postulated to play a role in destabilizing the fork, may be the basis for this effect. First, it was shown in the crystal structure of RecBCD that there is a 'pin' domain in RecC against which the fork is split enabling RecB and RecD to pull the opposite unwound ssDNA strands (*Singleton et al., 2004*). Second, it has been postulated that the 'arm' protrusion in RecB may play a direct role in destabilizing the duplex ahead of the translocating enzyme (analogous to the role suggested for auxiliary domain 2B of the PcrA helicase) (*Dillingham and Kowalczykowski, 2008*; *Velankar et al., 1999*) and a recent work showed that an arm deletion results in RecBCD's inability to unwind DNA (*Lohman and Fazio, 2018*; *Simon et al., 2016*). In both cases, the synergy between the subunits is expressed in the separation between the domain

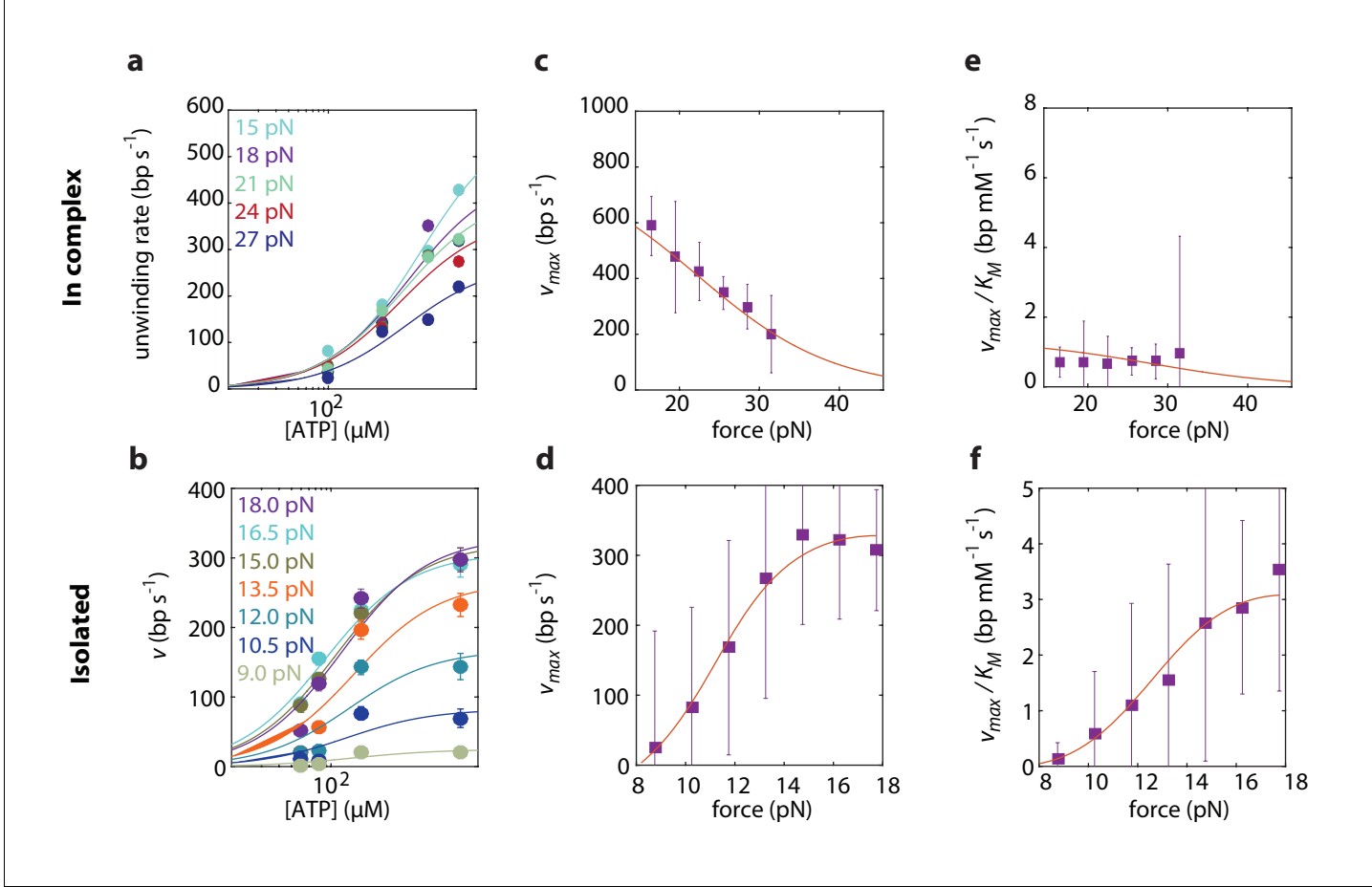

**Figure 4.** Force dependent Michaelis-Menten curves for RecD in complex (**a**) and isolated (**b**). Data shown as mean ± s.e.m. Lines through the data are the result of a global fit to a kinetic model of a Brownian ratchet before ATP binding (Equations S7 and S8 in *Supplementary file 2* and Equations S19 and S20 in *Supplementary file 3*). Different colors indicate different forces. (**c–d**). Force-dependent $v_{max}$ for RecD in complex (**c**) and isolated RecD (**d**). Discrete data points indicate $v_{max}$ as a result of fitting Michaelis-Menten curves to the data in a and b for each force separately (shown as fit result ± s. e.). Lines through the data indicate the results of the global fitting. (**e–f**) Force dependent $v_{max}/K_M$ for RecD in complex (**e**) and isolated (**f**). Discrete data points indicate $v_{max}/K_M$ as a result of fitting Michaelis-Menten curves to the data in a and b for each force separately (shown as fit result ± s.e.). Lines through the data indicate the results of the global fitting.

DOI: https://doi.org/10.7554/eLife.40836.009

The following figure supplements are available for figure 4:

**Figure supplement 1.** Global fits for different translocation mechanisms of in-complex RecD under force.
DOI: https://doi.org/10.7554/eLife.40836.010

**Figure supplement 2.** Global fits for different translocation mechanisms of isolated RecD under force.
DOI: https://doi.org/10.7554/eLife.40836.011

responsible for destabilizing the fork, and the fast translocase pulling the structurally unwound strands. Both features are needed for full activity by the complex.

Incidentally, the results from the global fitting can provide additional information on the mechanism of unwinding. First, they reveal that RecD's step size is ~3 bps, which is in accordance with previously reported step sizes characterized by RecBCD when RecD is the leading subunit (before the Chi sequence) (*Lucius et al., 2002*) but, surprisingly, not with the 1 bp step size reported for the related SF1 helicase UvrD (*Lee and Yang, 2006*). In addition, RecD's destabilization energy, $\Delta G_{RecD} \approx 0.6\ k_b T$, further demonstrates that RecD is a 'passive' enzyme, which is able to translocate only thanks to the thermal breathing of the fork.

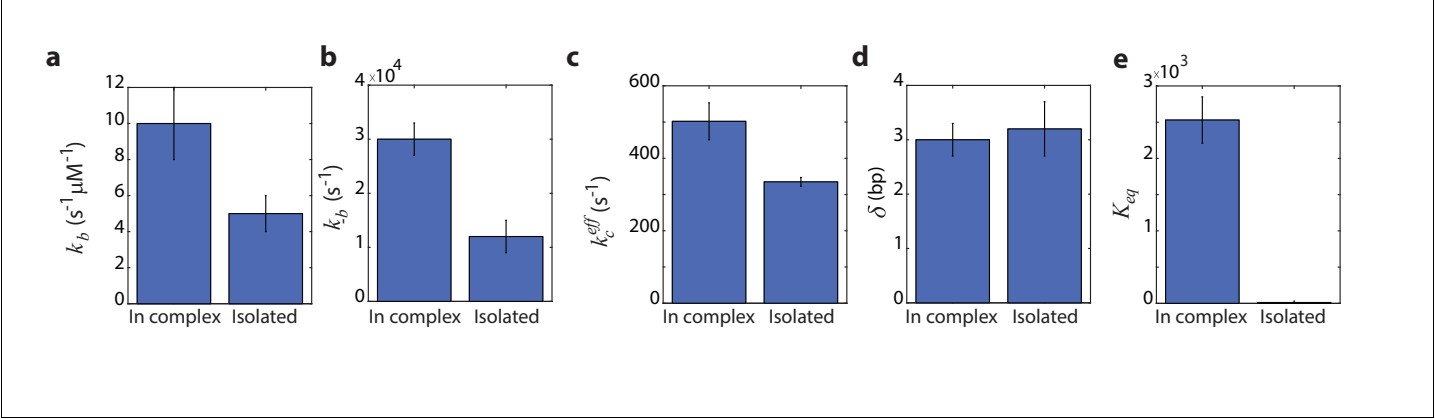

**Figure 5.** The global fitting parameters reveal a shift towards the post translocation state of RecD when in complex. (**a**) Forward ATP binding rate, $k_b$. (**b**) Backward binding rate, $k_{-b}$. (**c**) effective catalytic rate, $k_c^{eff}$. (**d**) Step size, $\delta$. (**f**) Equilibrium constant of translocation, shifted by the complex toward the post-translocation state.

DOI: https://doi.org/10.7554/eLife.40836.012

## Discussion

Processive helicases couple ATP hydrolysis with translocation on the DNA. Hence, their function takes place in a 2D mechano-chemical energy landscape that is affected by both [ATP] (a chemical reaction coordinate) and an external force (a mechanical reaction coordinate) (*Bustamante et al., 2004*; *Keller and Bustamante, 2000*). Hence, by studying how a helicase is affected by [ATP] and force, we can learn about the biochemical and mechanical transitions in its mechano-chemical cycle. However, for a heterotrimeric helicase, such as RecBCD, a comprehensive characterization of the mechano-chemistry presents additional challenges, since each of its helicase subunits is likely to play a different role in the generation of force. Moreover, although mutagenesis can be used to dissect the specific role of different units in a complex, it has its limitations too, since it requires to assume a priori the allosteric effect that one subunit has on the others. Hence, a method that can probe the activity of a single subunit, in the context of a complete complex, is of great interest. Here, we present such a method, and use it to probe the activity of the bacterial RecBCD and its subunits.

Using our novel single-molecule optical tweezers assay, we characterized the response of RecBCD and its subunits to an external force. Previous studies showed that RecBCD is able to overcome DNA-bound proteins without a significant effect on its velocity (*Finkelstein et al., 2010*; *Terakawa et al., 2017*), often pushing them thousands of base pairs before evicting them from DNA. A similar behavior was observed for a variety of DNA-binding proteins and also for nucleosomes, suggesting that RecBCD uses mechanical force to disrupt these obstacles in its way. However, these experiments cannot provide information of the magnitude of the force. Our measurements show that RecBCD is able to generate forces that reach up to ~40 pN, providing a mechanical basis for these previous observations. Notably, the forces measured here seem to be inconsistent with previous reports, where a maximal force of ~6 pN was reported for a biotinylated-RecBCD attached to a surface unwinding a DNA molecule under tension (*Perkins et al., 2004*). These differences may be related to the different chemical conditions at which the experiments were performed. The previous studies used a very low [ATP], aimed at maximizing the spatial resolution of the experiments. Under these conditions, the enzyme spends most of the time fluctuating between the pre- and post-translocation states, waiting for an ATP molecule that can 'lock' it in the post-translocation state and allow it to continue the catalytic cycle. These dynamic states (pre-translocation and post-translocation with no ATP bound) are likely characterized by weaker protein-DNA interactions and thus particularly sensitive to an applied force. Since the complex has an increased force-dependent probability of dissociation from these states, and it dwells longer in them when ATP is scarce, the net result is that at very low ATP concentrations RecBCD will be prone to dissociate with the application of a moderate force. Higher [ATP] allows the enzyme to minimize the time it spends in the force-sensitive states, thus reaching much higher forces. Another possible explanation

stems from the different geometry used in the experiments. It has been shown that due to the difference in unwinding rates by the subunits, a loop forms on one strand of the unwound DNA. The formation of the loop may have a significant effect in the biotinylated RecBCD setup previously reported, and might be the reason for RecBCD's slippage against forces larger than ~8 pN (*Perkins et al., 2004*). In our setup, where each subunit unwinds a different dsDNA segment, no loop formation takes place. Interestingly, if this is the reason for the differences observed, it may stress the potential regulatory role of the loop. Additional studies will be required to clarify this important point.

We characterized the role of the different subunits in the generation of force, by comparing RecD's activity when in complex to its activity as an isolated protein. Our findings indicate that the isolated RecD is a fast ssDNA translocase, but a very weak and poorly processive helicase, with a fork destabilizing energy of only $\sim 0.6 \pm 0.1 \ k_BT$. However, when complexed with the other subunits, RecD can efficiently and processively unwind DNA. This supports a model in which the helicase subunits function by a division of labor throughout the unwinding reaction. While the helicase subunits pull the DNA, they seem to play a very minor role in destabilizing the fork. Hence, to further clarify the mechanism by which RecB/RecC facilitate unwinding by RecD, we characterized the full mechano-chemical cycle of RecD, and how it is affected by the presence of the complex. Our measurements suggest that RecD unwinds the DNA as a Brownian ratchet that is rectified by ATP binding. Interestingly, a previous single-molecule study reported the existence of nanometer-scale conformational dynamics in RecBCD-DNA (*Carter et al., 2016*). These dynamics were observed also in the absence of ATP, but suppressed by ADP-BeF$_x$. The BR mechanism described here seems to be consistent with this previous report, and suggest that the conformational dynamics observed may represent the ratchet fluctuations, which exist independently of ATP, but are suppressed if the enzyme is locked in the post-translocation state by the non-hydrolyzable analog. The similar size of these dynamics (~4 bp) and the step size measured here (~3.3 bp) further support this interpretation. Notably, our results show that the effect of the additional units on RecD is expressed in shifting the ratchet equilibrium toward the post-translocation state. We speculate that this effect may be mediated by two structural motifs in the RecBCD complex. First, it has been shown that RecC has a pin that separates the complex, sending each strand into the RecB and RecD subunits (*Dillingham and Kowalczykowski, 2008*; *Singleton et al., 2004*). This results in RecD translocating on the single stranded DNA that has been unwound by RecC. Second, a recent work suggested that RecBCD can unwind duplex DNA processively in the absence of ssDNA translocation by the canonical motors (*Simon et al., 2016*). This was related to the arm protrusion in RecB that interacts with the double-stranded DNA ahead, which may also play a role in destabilizing the fork. Notably, the role of separate structural domains in destabilizing the duplex DNA is not a new proposal. In fact, a common feature of structural models for DNA unwinding by helicases is that melting occurs as the ssDNA translocase pulls the duplex DNA against a protein wedge or pin at the ss/dsDNA junction. For RecBCD, the structure of the full complex suggests that the pin is in RecC (*Singleton et al., 2004*). However, despite these structural insights, functional information was lacking. By comparing the activity of the RecD subunit, when it is part of a wild-type, complete RecBCD complex to its activity in isolation, our assays allow us to experimentally demonstrate a functional role for the presence of the other subunits.

Our assay has limitations: Normally, as a DNA molecule is unwound by RecBCD, two complementary strands are translocated by the two subunits, RecB and RecD. This means that RecC's pin, or any other structural element involved in unwinding, is accurately positioned to split these strands, for example by engaging at the DNA fork. In our assay, each of the helicase subunits (RecB and RecD) unwinds a separate dsDNA segment and therefore a structural element normally destabilizing the fork cannot be simultaneously destabilizing the two DNA segments. On the other hand, our experiments clearly demonstrate that RecBCD unwinds the DNA (*Figure 2a*), and that it does so by exploiting the unwinding activity of *both* helicases, RecB and RecD (*Figure 2b,c*). One possible way to reconcile these observations is that there is more than one structural element catalyzing unwinding, and that in our assay at least one of them assists the unwinding by RecB, while at least another one assist RecD. A different explanation, which we favor, is that RecBCD can fluctuate between two different conformations: one in which the pin, or other structural elements, are positioned to assist unwinding of the dsDNA in RecB, and therefore no unwinding takes place by RecD, and a conformation that allows unwinding of the DNA in RecD, but not the one in RecB. As a result, since the

subunits alternate between active and inactive states, their average velocity is slower than the velocity under normal operation, thus meaning that our assay underestimates the unwinding velocities. However, this possible underestimation does not affect our conclusions. First, the velocities measured are just multiplied by a constant factor: In the first possible scenario, this is the partial degree of destabilization offered by having only a subset of the destabilizing elements for each subunit. In the second scenario, this is the fraction of the time RecBCD spends in each of the conformations. Up to this numerical constant, the functional dependence of the velocity on the force as we measure truly represents the force response of the whole enzyme and its subunits. Specifically, for RecD, the underestimation of its velocity in the in-complex assay only strengthens the fact that at the zero-force extrapolation of *Figure 2b* (>400 bp/s) and *Figure 3c* (~0 bp/s) are in disagreement, and suggest the existence of a destabilizing element in the complex. Moreover, the modeling and global fitting are not affected, with the exception of a possible underestimation of the in-complex $K_{eq}$ in *Figure 2e*.

We recently reported the existence of additional, non-catalytic but functionally important ATP-binding sites in RecBCD (*Zananiri et al., 2017*). The requirement of both binding and dissociation of ATP from these sites during unwinding indicates that they serve to transfer ATP to the catalytic sites, thus increasing the total ATP flux at intermediate concentrations. It is possible that the existence of the auxiliary sites and the results presented here are related: In the case of a BR coupled to ATP binding, an opposing force such as an externally applied force or a DNA-binding protein presenting a roadblock for translocation, decreases the rate of reaching the post-translocation state, and therefore the rate of ATP binding. Hence, it is possible that the ATP auxiliary sites play a role in overcoming the effect of the force, by allowing binding via a parallel pathway.

## Materials and methods

### Reagents and purification of protein constructs

All chemicals and reagents were the highest purity commercially available. ATP was purchased from Roche Molecular Biochemicals (Indianapolis, IN). A molar equivalent of $MgCl_2$ was added to nucleotides immediately before use. Nucleotide concentrations were determined by absorbance using an extinction coefficient $\varepsilon_{259}$ of 15,400 $M^{-1}$ $cm^{-1}$. Unless otherwise specified, all experiments were conducted in RecBCD Buffer (RB: 20 mM MOPS pH 7.4, 2 mM $MgCl_2$, 1 mM DTT, 0.1 mM EDTA and, unless specified, 75 NaCl). Over-expression and purification of recombinant RecBCD was based on the method described by Roman *et. al.* (*Roman and Kowalczykowski, 1989b*). All steps of purification were carried out at 4°C, and contained RB at the indicated NaCl concertation, in addition to 1 mM PMSF, 1 mM Benzamidine. Four liters of *E. coli* cells expressing RecBCD were lysed using Microfluidizer, followed by centrifugation at 10,000 × g. The supernatant was further clarified by centrifugation at 100,000 × g and treated with Benzonase for two hours before initial purification by DEAE chromatography (a weak anion exchanger, to remove nucleic acids contaminants) using a linear NaCl gradient from 75 mM to 700 mM. RecBCD-containing DEAE fractions were eluted from a Q-sepharose column (a strong anion exchanger which highly selects for active RecBCD) using a linear NaCl gradient from 75 mM to 1 M. Fractions containing RecBCD were precipitated using $(NH_4)_2SO_4$ (45% saturation), and collected by centrifugation at 14,000 × g. Precipitated RecBCD was resuspended and loaded onto Superdex 200 equilibrated with RB, and the monodisperse peak containing the heterotrimer complex of RecBCD was collected. Fractions containing purified RecBCD were concentrated using an Amicon concentrator (50 kDa cutoff), aliquoted and flash frozen in liquid nitrogen before storage at −80°C. RecBCD concentration was determined using an extinction coefficient $\varepsilon_{280nm}$ of 4.2 × 10$^5$ $M^{-1}$ $cm^{-1}$ in Guanidine chloride. RecB$^{K29Q}$CD and RecBC were obtained by transforming the RecB$^{K29Q}$ ATPase mutant pPB800 (a gift from S. Kowalczykowski) and pPB700 (a gift from P. Bianco) plasmids, respectively, into the RecBD-null V330 strain, and following the same purification protocol as for WT RecBCD. RecBC concentration was determined using $\varepsilon_{ex,coeff.}$ of 3.7 × 10$^5$ $M^{-1}$ $cm^{-1}$.

The RecD gene flanked by HindIII and NdeI was obtained via PCR on the pPB800 plasmid with forward (5'-CTGATCGCATATGAAATTGCAAAAGCAATTACTGGAAGCTGTGGAG-3') and backward (5'-GCTGACTAAAGCTTTTATTCCCGTGAACTAAACAACGCCGCCA-3') primers (IDT), purified, cut and ligated into a HindIII and NdeI treated PET15b plasmid. Gene insertion was verified via

sequencing, then heat-shock transformed into BL21 bacteria. All purification procedures were carried out at 4°C. Similar to the purification from inclusion bodies by *Chen et al. (1997)*, eight liters of *E.coli* cells expressing Histidine tagged RecD (RecD for short) were lysed in Lysis Buffer (20 mM Tris-HCl pH 7.5, 500 mM NaCl, 1 mM Benzamidine and 1 mM PMSF) using the microfluidizer, and centrifuged for 10 min at 7000 × g. Pellets containing RecD in inclusion bodies were suspended in Resuspension Buffer (20 mM Tris-HCl pH 7.5, 500 mM NaCl, 6 M Guanidinium chloride, 5 mM Imidazole) and centrifuged for 45 min at 50,000 × g. The supernatant was applied to a pre-equilibrated (Equilibration buffer: Tris-HCl pH 7.5, 500 mM NaCl, 8 M Urea and 5 mM Imidazole) Nickel column (2 ml of HisPur, Thermo), and eluted with a stepwise gradient of imidazole (25–300 mM). Fractions containing denatured RecD where loaded onto Superdex 200 equilibrated with 20 mM Tris-HCl, pH 7.5, 500 mM NaCl and 6 M Urea. Fractions containing RecD, were concentrated using an Amicon concentrator (30 kDa cutoff) and gradient dialyzed into storage buffer (20 mM Tris-HCl, pH 8.0, 1 mM DTT, 0.05% Triton X-100, 500 mM NaCl, 20% glycerol). Then, it was aliquoted and flush frozen in liquid nitrogen before storage at −80°C. RecD concentration was determined using an extinction coefficient $\varepsilon_{280nm}$ of $4.8 \times 10^4$ $M^{-1}$ $cm^{-1}$ in Guanidine chloride.

## Molecular constructs for single-molecule experiments

We generated unwinding/translocation tracks of different lengths similarly to previously described methods (*Rudnizky et al., 2016*; *Rudnizky et al., 2018*). 600 and 4200 bp tracks were obtained using standard PCR reactions (*Supplementary file 5*, IDT), nicked using Nt.BbvCI for the Biotin-terminated track and Nb.BbvCI for the Digoxygenin-terminated one (enzymes from New England Biolabs), resulting in complementary 29-nucleotides, flanked with three nucleotides (5'-TGC-3'). For the symmetric geometry, the 600 biotin and digoxigenin tracks were mixed at equal molar ratios for DNA annealing, creating a ~1200 bp fragment. For the asymmetric geometries, 4200 bp handles were annealed to complementary purchased oligonucleotides with the opposite modification (*Supplementary file 5*, HPLC purified, IDT). This resulted in asymmetric handles with 4200 bps and ~ 35 nt single stranded DNA on opposite sides. All constructs were ligated to a ~ 250 dsDNA stem ('601' DNA) generated as previously described (*Rudnizky et al., 2016*).

## Optical tweezers

Experiments were performed in a custom-made double-trap optical tweezers apparatus (*Moffitt et al., 2006*), as previously described (*Malik et al., 2017b*; *Rudnizky et al., 2016*). Briefly, the beam from a 855 nm laser (TA PRO, Toptica) was coupled into a polarization-maintaining single-mode optical fiber. The collimated beam out of the fiber was split by a polarizing beam splitter (PBS) into two orthogonal polarizations, each directed into a mirror and combined again with a second BS. One of the mirrors is mounted on a nanometer scale mirror mount (Nano-MTA, Mad City Labs). A X2 telescope expands the beam, and also images the plane of the mirrors into the back focal plane of the focusing microscope objective (Nikon, Plan Apo VC 60X, NA/1.2). Two optical traps are formed at the objective's focal plane, each by a different polarization, and with a typical stiffness of 0.3–0.5 pN/nm. The light is collected by a second, identical objective, the two polarizations separated by a PBS, and imaged onto two Position Sensitive Detectors (First Sensor). The position of the beads relative to the center of the trap is determined by back focal plane interferometry (*Gittes and Schmidt, 1998*). Calibration of the setup was done by analysis of the thermal fluctuations of the trapped beads (*Berg-Sørensen and Flyvbjerg, 2004*), which were sampled at 100 kHz. Separate measurements confirmed that the beads remain in the optical traps' linear range for the entire experiment (*Figure 1—figure supplement 2*).

## Single-molecule experiments

The complete DNA construct was incubated for 15 min on ice with 0.9 µm polystyrene beads (Spherotech), coated with anti-Digoxygenin (anti-DIG). The reaction was then diluted 1000-fold in RB, with the addition of a 1:1 ratio of Mg·ATP, 0.05 mg/ml BSA, and an ATP regeneration system consisting of 7.5 mM Phosphocreatine and 0.05 mg/ml Creatine phosphokinase. Tether formation was performed in situ (inside the experimental chamber) by trapping an anti-DIG bead (bound by DNA) in one trap, trapping a 0.9 µm streptavidin-coated polystyrene beads in the second trap, and bringing the two beads into close proximity to allow binding of the biotin tag in the DNA to the streptavidin

in the bead. The laminar flow cell (Lumicks) had four channels: streptavidin beads pre-bound to the DNA construct, anti-digoxigenin beads, RB, and RB with the addition of RecBCD/RecD (25 nM). Single DNA tethers were verified in the buffer-only channel and then held at a tension of 5 pN and translocated to the RecBCD channel, until activity was observed as indicated by a rapid decrease in the extension and increase in the force. Data were digitized at a sampling rate $f_s$ = 2,500 Hz and saved to a disk. All further processing of the data was done with Matlab (Mathworks). The measured extension was transformed into contour lengths, in units of bp of dsDNA, using the extensible worm-like chain model with a persistence length of 40 nm and a stretching modulus of 1000 pN.

## Force-velocity curves

Force-time data were smoothed using a moving average filter to $f_c$ = 250 Hz. Instantaneous force velocity curves were then calculated as a linear fit in a 100 ms time window of the contour-time data, the velocity in the time window was taken as the mean velocity. Instantaneous force velocity data was further smoothed using a median filter one tenth of the data length. Smoothed force velocity curves were then quantized in the range of 0–60 pN with a bin size of 3 pN for RecBCD data and 2pN for isolated RecD. Force-velocity curves were then averaged over the ensemble of experiments.

## Derivation of Michaelis-Menten parameters for the different kinetic models

By combining two mechanism of translocation (BR and PS) with three potential locations for the translocation step within the chemical cycle (with/before ATP binding, with/before ATP hydrolysis, and with/before the release of the hydrolysis products), six potential kinetic schemes are formulated (*Supplementary file 1*). The derivation of the force-dependent Michaelis-Menten parameters for these models is based on the net rate theory (*Cleland, 1975*; *Dangkulwanich et al., 2013*), considering how force affects a translocation step in each assay: In the assay in Fig. 1a, where unwinding takes place against an opposing force, a translocation step $\left( R_n \underset{k_{-i,0}}{\overset{k_{+i,0}}{\rightleftharpoons}} R_{n+1} \right)$ is affected by force such that the forward rate will decrease by a factor of $k_{+i,0} \rightarrow k_{+i,0} \exp\left(-\frac{Fx^{\ddagger}}{k_b T}\right)$, where $x^{\ddagger}$ is the distance to transition state, while the backward rate will increase by a factor of $k_{-i,0} \rightarrow k_{-i,0} \exp\left(F(\delta - x^{\ddagger})/k_b T\right)$, where $\delta$ is the step size (*Bustamante et al., 2004*). Under rapid equilibrium conditions, the force will affect the equilibrium constant $K_i = k_{-i}/k_{+i}$ according to $K_i \rightarrow K_i \exp(F\delta/k_b T) \equiv K_i \exp(F/F_0)$. In the unwinding of a hairpin under tension setup (Figure 3a), the translocation step is modulated by force such that the forward rate is modulated by a factor $k_{+i,0} \rightarrow k_{i,0} P_{open}$, where $P_{open}$ is the probability that at least $n$ bp ahead are unwound, and $n$ is the helicase step size in bp. $P_{open}$ was calculated as previously reported (*Malik et al., 2017a*). Together, this results in separate predictions for the force-dependence of the Michaelis-Menten parameters for each of the models in *Supplementary file 1* and in each of the experimental assays. These predictions are summarized in *Supplementary file 2* and *3*.

Interestingly, in cases where $v_{max}$ decreases with force, all PS models lead to a simple relation:

$$v(F) = \frac{v_0}{1 + \exp\left[\frac{F - F_{1/2}}{F_0}\right]} \tag{1}$$

as obtained in earlier reports (*Abbondanzieri et al., 2005*). This expression is also valid for BR models, assuming that $x^{\ddagger} \approx \delta$ and that the translocation rates ($k_{\pm tr}$) are faster than all other rates dominating their corresponding exponents. *Equation 1* was used to fit force velocity curves in *Figure 2*.

## Global fitting of the mechano-chemical cycle

To elucidate which of the models best describes RecD's translocation mechanism, we used two sets of data of the form $v(F, [ATP])$. The first, $D_1$, measured in the 'unwinding under force' geometry (*Figure 1*) and the second, $D_2$, measure in the 'hairpin under tension' assay (*Figure 3*). Using the expressions for the expected force-dependent $v_{max}$ and $K_M$ it is possible, for any given model-dependent set of parameters, to calculate the expected velocity for the values of $F$ and $[ATP]$ at which the experimental data was taken, $M_1^i$ and $M_2^i$, where the superscript $i$ represents the specific model, and

the subscript 1,2 the experimental geometry. To find the parameters that maximize the fit of each model we minimized the sum of square errors (SSE) between the experimental data and the calculated values, that is minimizing $SSE_1 = \sum_{F,\,[ATP]}\left(D_1 - M_1^i\right)^2$ and $SSE_2 = \sum_{F,\,[ATP]}\left(D_2 - M_2^i\right)^2$ for each model over their relevant set of parameters, using a Global search optimization (*Ugray et al., 2007*) implemented in MATLAB. The model resulting in the lowest sum of minimized SSE was chosen to be the model best describing RecD's translocation mechanism. The goodness of the fits in *Figure 4—figure supplement 1* and *2* were calculated as $R^2 = 1 - SSE/SST$, where $SST$ is the total sum of squares. For illustration, Michaelis-Menten parameters were obtained from fitting Michaelis-Menten expressions separately for each force range (*Figure 4c–f*, data points) and are compared to the predictions of the selected model, with the parameters found (*Figure 4c–f*, lines).

## Acknowledgements

We thank Dr. Stephen C Kowalczykowski and Dr. Theetha Pavankumar (University of California, Davis, California) for the RecBCD expression system and their assistance with the purification protocol. We are grateful to Dr. Piero R Bianco (University at Buffalo, Buffalo, NY) for the RecBC null strain and RecBCD plasmids, and for his excellent advice.

## Additional information

### Funding

| Funder | Grant reference number | Author |
| --- | --- | --- |
| Israel Science Foundation | 1782/17 | Ariel Kaplan |
| Israeli Centers for Research Excellence | 1902/12 | Ariel Kaplan |
| Eliyhau Pen Research Fund | | Ariel Kaplan |
| Israel Science Foundation | 293/13 | Arnon Henn |

The funders had no role in study design, data collection and interpretation, or the decision to submit the work for publication.

### Author contributions

Rani Zananiri, Conceptualization, Software, Formal analysis, Investigation, Visualization, Methodology, Writing—original draft, Writing—review and editing; Omri Malik, Resources, Software, Methodology; Sergei Rudnizky, Vera Gaydar, Roman Kreiserman, Resources, Methodology; Arnon Henn, Conceptualization, Supervision, Funding acquisition, Methodology, Project administration, Writing—review and editing; Ariel Kaplan, Conceptualization, Resources, Formal analysis, Supervision, Funding acquisition, Methodology, Writing—original draft, Project administration, Writing—review and editing

### Author ORCIDs
Rani Zananiri ![ORCID] http://orcid.org/0000-0002-0094-4197
Arnon Henn ![ORCID] http://orcid.org/0000-0001-6360-7211
Ariel Kaplan ![ORCID] http://orcid.org/0000-0002-9731-6962

### Decision letter and Author response
Decision letter https://doi.org/10.7554/eLife.40836.025
Author response https://doi.org/10.7554/eLife.40836.026

## Additional files

### Supplementary files
• Supplementary file 1. All the possible mechano-chemical models.

DOI: https://doi.org/10.7554/eLife.40836.013

• Supplementary file 2. Force-dependent Michaelis-Menten parameters for the different models of unwinding, for experiments measuring activity against an opposing force.

DOI: https://doi.org/10.7554/eLife.40836.014

• Supplementary file 3. Force-dependent Michaelis-Menten parameters for the different models of unwinding, for experiments measuring unwinding of a hairpin under tension.

DOI: https://doi.org/10.7554/eLife.40836.015

• Supplementary file 4. Number of traces measured

DOI: https://doi.org/10.7554/eLife.40836.016

• Supplementary file 5. Oligonucleotides used for synthesizing DNA substrates for optical tweezers experiments.

DOI: https://doi.org/10.7554/eLife.40836.017

• Supplementary file 6. Results of fitting RecD force velocity curves to a Brownian ratchet before hydrolysis model, in complex and isolated.

DOI: https://doi.org/10.7554/eLife.40836.018

• Transparent reporting form

DOI: https://doi.org/10.7554/eLife.40836.019

## Data availability

All data generated during this study have been deposited in Dryad under accession code doi:10.5061/dryad.jb10510

The following dataset was generated:

| Author(s) | Year | Dataset title | Dataset URL | Database and Identifier |
|---|---|---|---|---|
| Zananiri R, Malik O, Rudnizky S, Gaydar V, Henn A, Kaplan A | 2018 | Data from: Synergy between RecBCD subunits is essential for efficient DNA unwinding | https://dx.doi.org/10.5061/dryad.jb10510 | Dryad, 10.5061/dryad.jb10510 |

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
