## [Decision Letter]

Thank you for submitting your article "Synergy between RecBCD subunits is essential for efficient DNA unwinding" for consideration by *eLife*. Your article has been reviewed by two peer reviewers, and the evaluation has been overseen by a Reviewing Editor and John Kuriyan as the Senior Editor. The following individual involved in review of your submission has agreed to reveal her identity: Maria Spies (Reviewer #2).

The reviewers have discussed the reviews with one another and the Reviewing Editor has drafted this decision to help you prepare a revised submission.

Summary:

The authors have developed a very clever optical-tweezer-based assay to monitor duplex unwinding/translocation and force generation by the two motor subunits within the holoenzyme of the RecBCD helicase/nuclease. The innovation is in the design of the substrate. They use a hairpin-unzippering assay, but without a loop at the end of the substrate that allows for RecBCD to load from a blunt end. The enzyme then unwinds the unloaded dsDNA of the hairpin until it encounters the base of the hairpin and starts to simultaneously unwind two dsDNA substrates due to the two helicases in RecBCD, one going 5'-3' and the other 3'-5'. By placing the base of the hairpin near one of the two beds, the motor activity of the RecB or RecD was investigated within the context of the intact RecBCD complex but where one of the motors had stalled by coming to the end of its substrate. RecD's activity while not in the RecBCD context was investigate in similar assay but where the enzyme unwinds the base of the hairpin, akin to pioneering helicase hairpin unwinding experiments by Bustamante and colleagues (Dumont, 2006). In both of these experiments, the force is free to vary as unwinding occurs (no force clamp), and the main data the authors collect is how the instantaneous unwinding rate changes as a function of force. They do this as a function of ATP concentration. This data is then compared against a number of kinetic models in order to determine the translocation mechanism of RecD, and how the other subunits modify that mechanism. The two most significant value added aspects of this work are (1) the ability to monitor the translocation and force generation by two RecBCD motors within the context of fully functional holoenzyme; and (2) a carefully elaborated model for the RecD Brownian ratchet mechanism. Overall the experiments are well designed and expertly executed. However, there are several major issues that need to be addressed before publication is considered.

Essential revisions:

First, there is great detail provided about the different kinetic modes developed to fit the force-velocity data, but there is not a corresponding attention to demonstrate the quality of the optical tweezer data. This information is critical to the manuscript because the authors are trying to distinguish different models based on their analysis of force-velocity. Hence force needs to be well measured. However, neither this manuscript nor a quick review through several of the preceding papers from this group seem to do a thorough presentation of the calibration results of their optical tweezers set up. There are several reasons to be concerned. First, the authors report measuring force of 50 pN with an optical tweezer stiffness of k = 0.3-0.5 pN/nm. This means the microsphere is displaced 100-150 nm from the trap center. To accurately measure force at this significant displacement, the authors need to demonstrate the linearity of their trap. There is clear evidence in the literature that if the authors are assuming F = -kx at such large displacement, there can be significant errors (Greenleaf, et al. 2005). Each tweezer setup is different, so it is critical that the authors demonstrate it for their setup. Second, they have two trapped microspheres quite close together (1200 bp = 400 nm) and the distance gets smaller as shown in Figure 1B. However, it is well known that there can be (1) cross-talk between both the two detector channels when using different polarizations for the same laser, (2) cross talk hydrodynamically between the microspheres, and (3) optical-induced effects of one trap being near another (see for example von Hansen, et al., 2012. Auto- and cross-power spectral analysis of dual trap optical tweezer experiments using Bayesian inference. Rev Sci Instrum and references therein). Another metric the authors should report is the persistence length they get for fitting their DNA to a WLC model and the other fitting parameters (such as stretch modulus).

Second, the manuscript seems to lack metrics presented to the reader for how the authors distinguished between competing kinetic models. The authors just state one model is the best. This is not acceptable as it doesn't allow for the reader or the reviewer to see how well the different models are distinguished. For an example of how to present model differentiation much better, see Supplementary Figure 1 of Larson, et al., 2012. Trigger loop dynamics mediate the balance between the transcriptional fidelity and speed of RNA polymerase II. PNAS. In the present manuscript, Figure 4 might be trying to do this, but it has several problems: (a) Curves in 4A and 4B are colored coded, but nowhere is it specified what the colors mean. (b) For 4C-F, the authors are fitting Brownian ratchet models to RecD Michalis Menten parameters. How did they calculate these parameters? It is never explicitly stated. The authors also mention that they did all of these studies as a function of ATP, so there should be a possibility of fitting everything to one global model (see for instance Figure 2 of Schnitzer, et al., 2000. Force production by single kinesin motors. Nature Cell Biology).

Third, the authors need to do a much better job discussing the nuances of force applied to the enzyme in their novel assay and the state of the DNA interacting with the RecBCD. In Figure 1A, the force is acting on the entrance to RecBCD and shearing the DNA, rather than across the enzyme and along the axis of the DNA that is seen in the Perkins et al., 2004 assay, where the enzyme is anchored to the surface. In Figure 1A, it might be quite conceivable that at 40 pN of tension within the substrate – well above the force needed to unzipper dsDNA – the DNA going into the front of RecBCD would both be ssDNA at the hole at the leading edge of RecBCD that normally admits dsDNA. Moreover, two segments of non-complementary ssDNA are being fed through the protein pore ahead of the pin. Hence, it is likely that the portion of RecB that interacts with the dsDNA ahead of the pin in the normal geometry may likely be interacting with ssDNA, if at all in this taut configuration. Also, the force in Figure 3B is assisting. The authors need to discuss both the benefits of their assay and limitations in drawing conclusions.

Fourth, the data shown in Figure 1B seems to be in contradiction to the force velocity curves shown in Figure 2. Note Figure 1B shows a significant reduction in velocity as the force is increased from ~6 pN to ~14 pN. It states it is a representative curve for RecBCD motion. Yet Figure 2 shows a flat force-velocity curve upto ~35-40 pN. Is there significant heterogeneity or different classes of response for ReBCD? Perhaps Figure 1B is at a lower ATP concentration, which would be interesting since it shows a different force-velocity response and as discussed in the Schnitzer kinesin paper, fitting the full force-velocity relation as a function of ATP concentration can yield interesting information and significantly constrain the model.

Fifth, the authors need to carefully discuss the geometric arrangement of the DNA-RecBCD complex. Structural studies of the RecBCD (X-ray crystallography and CryoEM from the Wigley lab) suggest several structural elements within RecB subunit are responsible for the duplex destabilization ahead of the RecBCD – the authors provide a very nice discussion of this. One would expect such duplex destabilization is essential for the unwinding mechanism. However, in the authors' experimental arrangement, only one of the two duplexes unwound by RecBCD may experience these duplex destabilizing elements. Would this underestimate the RecD activity? What are the possible structural/geometric arrangements with respect to the two duplexes being unwound?

Sixth, another puzzling observation is that if RecD is indeed the leading subunit, it should arrive at the junction ahead of RecB creating perhaps a loop. While RecB is catching up, RecD should start unwinding its duplex at its rate. After RecB catches up, both arms of the construct will be unwound. Hence, one would expect a region of the trace (at point 2 in Figure 1B) that will show a slower rate that is then increased. This slow rate should correspond to the RecD activity and the faster rate should be a sum of RecB and RecD-mediated unwinding of their respective arms. From the difference between the two rates, one, in principle, should be able to extract the rates for both subunits and compare them to the experiment with the entry site near the bead. Would this be possible, or is the length of the entry stem is two short?

---

## [Author Response]

Essential revisions:First, there is great detail provided about the different kinetic modes developed to fit the force-velocity data, but there is not a corresponding attention to demonstrate the quality of the optical tweezer data. This information is critical to the manuscript because the authors are trying to distinguish different models based on their analysis of force-velocity. Hence force needs to be well measured. However, neither this manuscript nor a quick review through several of the preceding papers from this group seem to do a thorough presentation of the calibration results of their optical tweezers set up. There are several reasons to be concerned. First, the authors report measuring force of 50 pN with an optical tweezer stiffness of k = 0.3-0.5 pN/nm. This means the microsphere is displaced 100-150 nm from the trap center. To accurately measure force at this significant displacement, the authors need to demonstrate the linearity of their trap. There is clear evidence in the literature that if the authors are assuming F = -kx at such large displacement, there can be significant errors (Greenleaf, et al. 2005). Each tweezer setup is different, so it is critical that the authors demonstrate it for their setup.

We agree on the importance of assessing the accuracy of our measurements, and have performed new experiments to better characterize it. In particular, to check the linearity of our traps in the relevant bead-displacement range, we performed an additional experiment presented in the new Figure 1—figure supplement 2.

The normalized shape of the trapping potential in general, and the size of its linear range in particular, depend on the size of the bead, its refractive index, the laser waist size and its wavelength. Importantly, the power of the laser affects the depth of the potential and the traps stiffness, but not the normalized shape. Hence, is it possible to estimate the linearity range of a high laser power trap by measuring it at low power. This fact was used in the experiment described in Figure 1—figure supplement 2A: Two similar beads were independently trapped (i.e. with no tether between them) in two traps that were adjusted (using the laser power) to have different spring-constants: One “strong” (k ~ 0.75 pN/nm) and one “weak” (k ~ 0.2 pN/nm). Then, using our laminar flow cell, we introduced a control flow increasing the fluid velocity every ~ 2s. The flow creates an identical force on both beads according to Stokes’ law (Figure 1—figure supplement 2B), up to small differences in size between the beads (<10% according to the manufacturer) which are seen as slight differences in the slope of the force vs. time (or fluid velocity) data. For the strong trap, this force produces a relatively small displacement of the bead (~ 33 nm for forces of 25 pN), and the measurement can thus be assumed to be in the linear range, and a reliable measurement of the force. For this same force, the displacement of the bead in the weak trap is much larger, up to ~125 nm, the range where we would like to check the linearity.

Hence, the linear dependence between the force (measured with the strong trap) and the displacement of the bead in the weak trap (Figure 1—figure supplement 2C) is a demonstration of the linearity of the latter in the relevant experimental range, and a confirmation of the accuracy of our force measurements. A complementary and identical experiment where the roles of the traps were reversed (not shown) demonstrated the linearity of both traps.

Second, they have two trapped microspheres quite close together (1200 bp = 400 nm) and the distance gets smaller as shown in Figure 1B. However, it is well known that there can be (1) cross-talk between both the two detector channels when using different polarizations for the same laser, (2) cross talk hydrodynamically between the microspheres, and (3) optical-induced effects of one trap being near another (see for example von Hansen, et al., 2012. Auto- and cross-power spectral analysis of dual trap optical tweezer experiments using Bayesian inference. Rev Sci Instrum and references therein).

It is indeed well known that there are optical and hydrodynamic disturbances in the measured signal as the two traps get closer. However, we are convinced that these disturbances do not have a significant effect in our data. To demonstrate that, we performed an additional experiment, described below.

Using two identical beads we recorded the measured forces as the beads approach each other as a function of the voltage applied to the piezo-controlled steerable mirror. The recorded pattern, with wavelength-scale periodicity, is the result of the effects described above. Next, we modified our experimental setup in the following way: Instead of using a single laser (TA-PRO, Toptica, λ = 855.2 nm) and setting the laser’s polarization at 45° so that a Polarizing Beam Splitter (PBS) separates the beam into two polarizations that form two traps, we rotated the laser’s polarization so that it creates a single trap, and introduced a second laser (TA-PRO, Toptica), of similar but not identical wavelength (λ = 852.2 nm) and orthogonal polarization, in order to form the second trap (panels A, B in the Author response image 1).

Importantly, the setup remained identical with the exception of the lasers’ source. As the two lasers are mutually incoherent, no interference is expected. Indeed, when we repeated the measurements with two beads trapped by two traps that were produced by different lasers, we measured a practically flat force almost all the way to the point of contact between the beads. This demonstrates that the main potential disturbance in the measurement results from interference between the traps.

Moreover, this experiment allows also us to quantify the magnitude of the disturbance to our force measurements. The beads used in our experiment make contact at a mirror voltage of ~2.6 V. Based on the calibration of the mirror, the 400 nm of the tether correspond to an additional 0.54 V in the mirror’s voltage. Hence, our experiments “start” at V~3.14 V. Assuming that, when the force reaches 50 pN, the beads are displaced by ~ 100 nm from the center of the trap (i.e. a total change of -200 nm in the bead-bead distance), our experiment measuring the activity of RecBCD is equivalent to moving on the blue curve, to the left by -0.27 V. As seen in the Author response image 1 (red arrow) the error in the force measurement acquired by a single trace can be estimated as only ± 0.5 pN in this range. Moreover, as the precise voltage of contact, and hence the range of the experiment, is a sensitive function of the size of the beads (which vary by 10% according to the manufacturer’s specifications; equivalent to ~90 nm or 0.12 V position change in Author response image 1), different experimental realizations sample a slightly different region of the interference curve (see the pink arrows in Author response image 1, for example) and acquire different errors as a function of force. This means that the ± 0.5 pN error range of a single trace is incoherently averaged to a lower value when

calculating the mean over the whole set of traces. In summary, although the interference between the traps likely adds to the experimental uncertainty, it is a very small effect at the level of ensemble-averaged force measurements.

**Author response image 1. respfig1:** Effect of the trap-trap interference at short distances. A. In our single-laser, dual-trap optical tweezers setup the two traps are created by splitting a single laser beam into two orthogonally polarized beams, using a polarizing beam splitter (PBS). After being reflected by steerable and fixed mirrors, respectively, the two beams are combined by a second PBS and directed to the focusing objective. Combined with differential detection, this setup offers a high degree of stability, since laser fluctuations are common to both traps and therefore “rejected”. However, the use of the orthogonal polarizations of a single laser with a high numerical objective is known to result in interference between the beams, and thus spurious position-dependent force signals. B. To assess the magnitude of this effect, we introduced a second laser, thus producing the traps by two mutually incoherent beams. C. Measurements with a single laser (blue) display a characteristic wavelength-scale modulation as the traps are brought into proximity by reducing the voltage on the steerable mirror, while two-lasers measurements are devoid of this modulation. The error in the force signal in the relevant position range (red arrow) is estimated as ± 0.5 pN for an individual trace. Due to the variability in bead size, there is some variability in the absolute range probed by different experiments (examples are shown in pink arrows), and thus the errors in *average* forces, as reported in the manuscript, are even smaller and have no significant effect in our measurements.

Another metric the authors should report is the persistence length they get for fitting their DNA to a WLC model and the other fitting parameters (such as stretch modulus).

These parameters have been added to the Materials and methods section.

Second, the manuscript seems to lack metrics presented to the reader for how the authors distinguished between competing kinetic models. The authors just state one model is the best. This is not acceptable as it doesn't allow for the reader or the reviewer to see how well the different models are distinguished. For an example of how to present model differentiation much better, see Supplementary Figure 1 of Larson, et al., 2012. Trigger loop dynamics mediate the balance between the transcriptional fidelity and speed of RNA polymerase II. PNAS. In the present manuscript, Figure 4 might be trying to do this, but it has several problems: (a) Curves in 4A and 4B are colored coded, but nowhere is it specified what the colors mean. (b) For 4C-F, the authors are fitting Brownian ratchet models to RecD Michalis Menten parameters. How did they calculate these parameters? It is never explicitly stated.

We thank the reviewer for stressing this point; the comparison between the models was indeed not described in enough detail and we have now significantly expanded and improved it. First, we have added a new section to the Materials and methods describing the whole fitting procedure (“Global fitting of the mechano-chemical cycle”). Next, we have added the missing color coding for the curves in 4A and 4B and incorporated into the caption of Figure 4 the details on how the Michaelis Menten parameters were calculated.

Finally, following the reviewer’s suggestion, we are now presenting in the new Figure 4—figure supplements 1 and 2, the results from fitting the different models to the data, similarly to the presentation by Larson and coworkers (Larson et al., 2012).

The authors also mention that they did all of these studies as a function of ATP, so there should be a possibility of fitting everything to one global model (see for instance Figure 2 of Schnitzer, et al., 2000. Force production by single kinesin motors. Nature Cell Biology).

We agree with the reviewers that fitting data taken at different forces and different [ATP] to a global model is very informative. In fact, this is exactly what we did in our paper. This was apparently not clear enough. We expect that the new section at the Materials and methods (see point above) describing the whole fitting procedure, the new Figure 4—figure supplements 1 and 2 describing the fitting results of all the kinetic models, and the improvement of the caption of Figure 4, have made this point clearer. To further clarify it, we have tried to improve our presentation. Please see the third paragraph of the subsection “RecBC stimulates RecD unwinding, by destabilizing the duplex DNA”.

Third, the authors need to do a much better job discussing the nuances of force applied to the enzyme in their novel assay and the state of the DNA interacting with the RecBCD. In Figure 1A, the force is acting on the entrance to RecBCD and shearing the DNA, rather than across the enzyme and along the axis of the DNA that is seen in the Perkins et al., 2004 assay, where the enzyme is anchored to the surface. In Figure 1A, it might be quite conceivable that at 40 pN of tension within the substrate – well above the force needed to unzipper dsDNA – the DNA going into the front of RecBCD would both be ssDNA at the hole at the leading edge of RecBCD that normally admits dsDNA.

The geometry of our experiments is clearly different that the one used by the Block and Perkins labs (Carter et al., 2016; Perkins et al., 2004). It is also important to notice that the effect of force is different in the initial phase, and the subsequent one (Please see the new Figure 1—figure supplement 1): Initially, RecBCD binds the DNA blunt end and unwinds the “stem”. During this phase, the force is in an “unzipping” configuration. However, the force is acting on the “fork” downstream from the binding site and is always lower than the force required to unzip the DNA. Hence, it does not have any effect on the DNA substrate or the enzyme’s activity. (And even if it did, since no extension changes take place at this phase nothing is measured. This is an invisible phase.). Then, as each subunit engages a DNA strand in a different “track” and starts to translocate, reducing the total extension (this is the measurement phase), the force becomes an opposing one, functioning as an inhibitor of translocation for both subunits, a fact that is exploited in our paper to study the mechanochemistry of the enzyme. To clarify how the force acts on the complex, we have now included a new Figure 1—figure supplement 1. Please see also subsection “Optical tweezers can monitor the force response of individual subunits in a native, full RecBCD complex”, first paragraph.

Next, the reviewer is concerned that the high forces reached during the measurement phase may affect the integrity of the DNA. However, although 40 pN is indeed above the force required to unzip the DNA, no unzipping is possible in the experimental geometry at the measurement phase (i.e. as the subunits translocate on the tracks). The DNA ahead of each of RecBCD’s subunits is under high force, but this is a stretching of shearing force, not an unzipping one (see Figure 1—figure supplement 1). Numerous previous studies have shown that it is possible to apply high stretching or shearing forces without affecting the dsDNA structure. For example, previous experiments combining optical tweezers with fluorescence imaging revealed that it is possible to stretch a long dsDNA molecule all the way to the overstretching transition at 65 pN without inducing any melting of the DNA (King et al., 2013; Mameren et al., 2009). In fact, even a short piece of duplex nucleic acid, ~35 bp of length, can sustain 40 pN of shearing force, as demonstrated, for example, by the use of a 35 bp DNA/RNA hybrid as a “handle” to unfold a riboswitch (Anthony et al., 2012; Duesterberg et al., 2015).

Moreover, two segments of non-complementary ssDNA are being fed through the protein pore ahead of the pin. Hence, it is likely that the portion of RecB that interacts with the dsDNA ahead of the pin in the normal geometry may likely be interacting with ssDNA, if at all in this taut configuration. The authors need to discuss both the benefits of their assay and limitations in drawing conclusions.

It is true that our experiment differs from the normal operation of RecBCD in that normally the strands translocated by RecB and RecD are complementary, while in our experiment they are not. However, to the best of our knowledge, the complementarity between the sequences has not been shown to play a role in the activity of RecBCD. In fact, RecB and RecD have been shown to function at different velocities, and therefore at any given point in time the sequences being translocated are non‐complementary.

Nonetheless, the reviewers are right in stressing the importance of explaining better the possible configuration of the RecBCD/DNA complex, and the limitations of our assay. Please see also our reply to the penultimate point.

Also, the force in Figure 3B is assisting.

Indeed, while the force opposes translocation in the assay of Figure 1, its assists translocation in the assay described in Figure 3. Moreover, while in Figure 1 the effect of the force is direct, and the force inhibits the motion forward because an additional mechanical work is required to translocate, in Figure 3 the effect of the force is indirect, with the assisting effect achieved by destabilizing the DNA fork ahead. The different “sign” of the force on the translocation (i.e. opposing or assisting), as well as the different functional dependence of the translocation step on the force, were described in the Supplementary Discussion in our original submission, and now (to comply with *eLife’s* format requirements) were incorporated into the Materials and methods. These considerations are an integral part of the models for translocation under force for each one of the geometries, as described by Supplementary files 2 and 3.

Fourth, the data shown in Figure 1B seems to be in contradiction to the force velocity curves shown in Figure 2. Note Figure 1B shows a significant reduction in velocity as the force is increased from ~6 pN to ~14 pN. It states it is a representative curve for RecBCD motion. Yet Figure 2 shows a flat force-velocity curve upto ~35-40 pN. Is there significant heterogeneity or different classes of response for ReBCD? Perhaps Figure 1B is at a lower ATP concentration, which would be interesting since it shows a different force-velocity response and as discussed in the Schnitzer kinesin paper, fitting the full force-velocity relation as a function of ATP concentration can yield interesting information and significantly constrain the model.

The trace previously shown in Figure 1B indeed corresponds to a lower ATP concentration (350 μM). For the sake of comparison, we have added another representative curve, from a higher ATP concentration of 2 mM ATP. Next, as is the case in most single-molecule studies, there is a certain amount of heterogeneity between the traces even under identical conditions. All the conclusions in the paper are drawn from averages over many traces. The heterogeneity is reflected in the averages’ uncertainty, i.e. the standard error of the mean, as shown in Figures 2 and 3.

Finally, please see our fifth reply above concerning fitting the full velocity vs. force and [ATP] data to a global model.

Fifth, the authors need to carefully discuss the geometric arrangement of the DNA-RecBCD complex. Structural studies of the RecBCD (X-ray crystallography and CryoEM from the Wigley lab) suggest several structural elements within RecB subunit are responsible for the duplex destabilization ahead of the RecBCD – the authors provide a very nice discussion of this. One would expect such duplex destabilization is essential for the unwinding mechanism. However, in the authors' experimental arrangement, only one of the two duplexes unwound by RecBCD may experience these duplex destabilizing elements. Would this underestimate the RecD activity? What are the possible structural/geometric arrangements with respect to the two duplexes being unwound?

The reviewers rise an important point, that we fail to discuss in our original submitted manuscript. Normally, as a DNA molecule is unwound, the two released complementary strands are translocated by the two subunits, RecB and RecD. This means that RecC’s pin, or any other structural element involved in unwinding, is accurately positioned to split these strands, e.g. by engaging directly with the fork-junction of the duplex DNA. In our assay, each of the helicase subunits (RecB and RecD) is unwinding a separate dsDNA segment and therefore a structural element essential for destabilizing the fork cannot be simultaneously destabilizing the fork of the dsDNA unwound by RecB and that or the DNA unwound by RecC. On the other hand, our experiments clearly demonstrate that RecBCD unwinds the DNA (Figure 2A), and that it does so by exploiting the unwinding activity of both helicases, RecB and RecD (Figure 2B, C).

One possible way to reconcile these observations is that there is more than one structural element contributing to the splitting of the DNA or stabilizing the unwound strands from reannealing, and that in our assay one such a motif (at least) assists the unwinding by RecB, while at least another unique motif assists RecD.

A different explanation, which we favor, is that RecBCD can fluctuate between two different conformations: one in which the pin, or other structural elements, are positioned to assist unwinding of the dsDNA in RecB, and therefore no unwinding takes place by RecD, and a conformation that allows unwinding of the DNA in RecD, but not the one in RecB. As a result, since the subunits alternate between active and inactive states, their average velocity is slower than the velocity under normal operation, thus meaning that our assay underestimates the unwinding velocities. However, this possible underestimation does not affect our conclusions. First, the velocities measured are just multiplied by a constant factor: In the first (possible) scenario, this is the partial degree of destabilization offered by having only a subset of the destabilizing elements for each subunit. In the second scenario, this is the fraction of the time RecBCD spends in each of the conformations. Up to this numerical constant, the functional dependence of the velocity on the force as we measure truly represents the force response of the whole enzyme and its subunits. Specifically, for RecD, the underestimation of its velocity in the in-complex assay only strengthens the fact that at the zero-force extrapolation of Figure 2B (>400 bp/s) and Figure 3C (~0 bp/s) are in disagreement, and suggest the existence of a destabilizing element in the complex. Moreover, the modeling is not affected, with the exception of a possible underestimation of the in-complex 𝐾*_eq_* in Figure 5E.

We have added a new paragraph about this point to the Discussion (fourth paragraph).

Sixth, another puzzling observation is that if RecD is indeed the leading subunit, it should arrive at the junction ahead of RecB creating perhaps a loop. While RecB is catching up, RecD should start unwinding its duplex at its rate. After RecB catches up, both arms of the construct will be unwound. Hence, one would expect a region of the trace (at point 2 in Figure 1B) that will show a slower rate that is then increased. This slow rate should correspond to the RecD activity and the faster rate should be a sum of RecB and RecD-mediated unwinding of their respective arms. From the difference between the two rates, one, in principle, should be able to extract the rates for both subunits and compare them to the experiment with the entry site near the bead. Would this be possible, or is the length of the entry stem is two short?

During the initial phase the two subunits of RecBCD are translocating on the two strands of the “stem” portion. Since this translocation doesn’t affect the end-to-end extension of the tether along the axis connecting the beads, no extension or force change is observed.

Extension changes are observed only as the stem is completely unwound and the different subutnis begin unwinding the tracks. The reviewers rise an interesting point here: If one subunit is faster than the other, it will reach its track first. And if that is the case, shouldn’t we expect to see a phase in the activity where only one subunit’s activity (the faster’s) is evident, followed by a second phase where we see both subunits’ activity?

We believe our data rules out this possibility, for the following reason. If the translocation velocity of the subunits is different a ssDNA segment will accumulate ahead of the slow subunit, forming a loop (as previously shown (Taylor and Smith, 2003)). This is schematically depicted in Author response image 2, at point 2. However, since the whole tether is under force, when the faster subunit reaches the junction (point 3) there is nothing which prevents the ssDNA in the loop to stretch, producing a sudden increase in the total tether length (point 4), that will then be followed by shortening of the tether as the subunits translocate on opposite directions. This phonotype is further described in panel B of Author response image 2.

Such sudden extension increase that precedes the activity was never observed in our data, ruling out the formation of a loop, or at least one of a significant size to be detected with our spatial and temporal resolution. This is perhaps not completely unexpected, since the velocity of the individual subunits as they unwind the stem, as estimated from Figure 2B and C by extrapolating to zero force, is not significantly different in our setup to facilitate the formation of a significant loop during the unwinding of the ~250 bp stem.

**Author response image 2. respfig2:**